# HIV viral load assays when used with whole blood perform well as a diagnostic assay for infants

**Charles Kiyaga[1]\*, Youyi Fong[2], Christopher Okiira[3], Grace Esther Kushemererwa[1], Ismail Kayongo[1], Iga Tadeo[1], Christine Namulindwa[1], Victor Bigira[3], Isaac Ssewanyana[1], Trevor Peter[4], Meg Doherty[5], Jilian A. Sacks[4], Lara Vojnov[5]**

**1** Central Public Health Laboratory, Kampala, Uganda, **2** Fred Hutchinson Cancer Research Center, Seattle, WA, United States of America, **3** Uganda National Health Laboratory Services, Kampala, Uganda, **4** Clinton Health Access Initiative, Boston, MA, United States of America, **5** World Health Organization, Geneva, Switzerland

\* ckiyaga@gmail.com

**Data Availability Statement:** All relevant data are within the manuscript.

**Funding:** DFID funded this study. The funder had no role in study design, data collection and

## Abstract

### Objective

Over the past several years, only approximately 50% of HIV-exposed infants received an early infant diagnosis test within the first two months of life. While high attrition and mortality account for some of the shortcomings in identifying HIV-infected infants early and putting them on life-saving treatment, fragmented and challenging laboratory systems are an added barrier. We sought to determine the accuracy of using HIV viral load assays for infant diagnosis of HIV.

### Methods

We enrolled 866 Ugandan infants between March–April 2018 for this study after initial laboratory diagnosis. The median age was seven months, while 33% of infants were less than three months of age. Study testing was done using either the Roche or Abbott molecular technologies at the Central Public Health Laboratory. Dried blood spot samples were prepared according to manufacturer-recommended protocols for both the qualitative and quantitative assays. Viral load test samples for the Roche assay were processed using two different buffers: phosphate-buffered saline (PBS: free virus elution viral load protocol [FVE]) and Sample Pre-Extraction Reagent (SPEX: qualitative buffer). Dried blood spot samples were processed for both assays on the Abbott using the manufacturer's standard infant diagnosis protocol. All infants received a qualitative test for clinical management and additional paired quantitative tests.

### Results

858 infants were included in the analysis, of which 50% were female. Over 75% of mothers received antiretroviral therapy, while approximately 65% of infants received infant prophylaxis. The Roche SPEX and Abbott technologies had high sensitivity (>95%) and specificity (>98%). The Roche FVE had lower sensitivity (85%) and viral load values.

analysis, decision to publish, or preparation of the manuscript.

**Competing interests:** The authors have declared that no competing interests exist.

## Conclusions

To simplify and streamline laboratory practices, HIV viral load may be used to diagnose HIV infection in infants, particularly using the Roche SPEX and Abbott technologies.

## Introduction

Infant diagnosis testing has expanded since 2010 with approximately 1.6 million tests performed in 2018 [1]. While it was estimated that 1.3 million HIV-exposed infants were in need of a diagnostic test in 2018, unfortunately, only 59% received a test within the first two months of life [2]. Further, in the same year less than 55% of children 0–14 years of age were accessing life-saving antiretroviral therapy [3]. Specifically, in Uganda, even though 93% of pregnant women living with HIV received antiretroviral therapy, only 45% of HIV-exposed infants received an early infant diagnosis test within the first two months of life [2]. Approximately 7,500 infants in Uganda became infected with HIV in 2018, while 66% of children aged 0–14 years of age received antiretroviral therapy [3]. To improve access to testing and necessary treatment, adjustments and efficiencies within the infant diagnosis system may be necessary [4, 5].

Challenges to the infant diagnosis system persist in high HIV burden countries, including fragile transport networks, sample collection and reagent stock outs, low volumes, and the need for sample batching that result in long test turnaround times, high prices, and fragmented procurement [4, 5]. Distinct workflow and processing stations are often implemented in laboratories to separate qualitative and quantitative procedures. The small, often-termed "orphan" market size of infant diagnosis, as well as unpredictable procurement practices can challenge manufacturer processes and cause availability delays [5]. Viral load testing, on the contrary, observes significant yearly volumes [6] and therefore more consistent procurement practices and lower prices. While same-day point-of-care testing is strongly recommended as the preferred testing approach by the WHO [7–10], when continuing to test using laboratory-based testing modalities, consolidating and simplifying infant testing using viral load assays through leveraging the viral load successes may reap significant benefits. Countries may access lower viral load prices and could remove the necessity to batch samples thus reducing delays, unify and simplify procurement, and consolidate volumes to ensure consistent reagent supply.

Previous studies have suggested that quantitative testing could be used to diagnose infants with HIV infection [11–13]. In one study, quantitative RNA testing was used for diagnosis in 156 HIV-exposed, non-breastfed infants less than six months of age, and no differences were observed in diagnostic accuracy compared to qualitative, DNA testing [12]. Further, a cohort of 96 infants were tested using HIV-1 RNA and DNA molecular assays using plasma samples and similarly no accuracy differences were found [13]. Based on these data, both WHO and US guidelines indicate that virologic assays that directly detect HIV (DNA, RNA, TNA, or p24) can be used to diagnose HIV infection in infants and children younger than 18 months of age, when serological assays cannot be reliably used [14–17]. Furthermore, US guidelines suggest that RNA molecular assays may be preferable for known maternal non-subtype B virus detection [14].

A large focus in the global community has been to prioritize qualitative HIV DNA assays for EID testing; however, early evidence suggests that RNA assays may be comparable [11–13]. Some additional concerns exist suggesting that maternal antiretroviral treatment access through Option B+ and *Treat All* policies, as well as provision of infant prophylaxis could

reduce levels of viremia in HIV-infected infants to undetectable levels, potentially requiring DNA-specific assays. We, therefore, conducted this study to better understand if HIV viral load quantitative assays can be used to diagnose HIV infection in infants by applying a qualitative interpretation of results.

## Materials and methods

This was a blinded, cross-sectional, prospective study to investigate the diagnostic accuracy of laboratory-based viral load quantitative assays to determine HIV infection compared to laboratory-based, qualitative infant diagnosis assays. All testing occurred at the Central Public Health Laboratory in Kampala, Uganda using remnant samples from routinely collected dried blood spot samples. Samples were received in the laboratory through the national infant diagnosis system from any health care facility in the country submitting a clinical sample from an HIV-exposed infant less than 18 months of age for routine diagnosis. Sample receipt, processing, and testing occurred between March and August 2018. All clinical samples were tested using the Roche COBAS AmpliPrep/COBAS TaqMan HIV-1 Qualitative Test, v2.0 (total nucleic acid detected)–these results were provided to the health care facility, health care workers, and caregivers to manage the infant's care. Samples were purposefully selected in that all consecutively collected positive samples and an equal number of randomly selected negative samples were included and blindly tested each week until the target sample size was met. Most (179) of the negative samples were used for both the Roche COBAS AmpliPrep/COBAS TaqMan HIV-1 Test, v2.0 and Abbott RealTime HIV-1 viral load assays (RNA only detected); however, 70 additional consecutive negative samples were collected for testing using the Abbott viral load assay, as the original samples were insufficient for testing with both assays. Separate sets of consecutively collected positive samples were used for the two technologies (Roche COBAS AmpliPrep/COBAS TaqMan HIV-1 Test, v2.0 and Abbott RealTime HIV-1 viral load), because the majority of positive samples did not have sufficient remaining spots available as all positive samples in routine clinical care are repeat tested in the laboratory prior to result dispatch.

Demographic and clinical data were collected from each patient using routine national requisition forms, including age, sex, maternal treatment, infant prophylaxis, and breastfeeding status. The cycle threshold of both qualitative and quantitative assays were captured as well as the qualitative result (detected or not detected) and viral load result from the quantitative assay.

Dried blood spot preparation and testing for the qualitative assays were conducted as previously described for the Roche COBAS AmpliPrep/COBAS TaqMan HIV-1 Qualitative Test, v2.0 [18]. Dried blood spots were prepared in two ways for the Roche COBAS AmpliPrep/COBAS TaqMan HIV-1 Test v2.0, using SPEX and PBS (free virus elution: FVE protocol) buffers [18, 19]. In brief, one spot was cut out using a pair of scissors or 12mm circular punch, transferred with forceps to an S-tube and 1100 ul of Sample Pre-Extraction Reagent (SPEX) was added; the tubes were incubated in a thermomixer at 56˚C and shaken at 1000 rpm for ten minutes before being loaded on to the sample rack for testing. For the COBAS AmpliPrep/COBAS TaqMan HIV-1 Qualitative Test, v2.0 using the FVE protocol, one spot was cut out using a pair of scissors or 12 mm circular punch, transferred with forceps to an S-tube and 1000 ul of calcium- and magnesium-free Phosphate buffered saline (PBS) buffer added; the tubes were incubated at room temperature for at least 30 minutes or overnight. The tubes were gently tapped at the bottom to homogenize the solution before being loaded on to the sample rack for testing. Dried blood spots for the Abbott RealTime HIV-1 Viral Load assay were prepared similarly to those prepared for the Abbott RealTime HIV-1 Qualitative assay [20]. In brief, one spot was punched from the card using a sterile pipet tip, placed in a tube, and 1300

ul of mSample Preparation System buffer added; the tubes were manually swirled to ensure the spot was fully submerged, and incubated in a thermomixer at 55˚C for 30 minutes. Tubes were then manually swirled again before being transferred directly to the sample rack for testing. Alternatively, as a sub-analysis to determine if a different sample preparation might improve performance, we also processed a separate set of samples using a modified dried blood spot sample preparation protocol, in which two spots were submerged in 1500 ul of mDBS buffer (all other steps remaining consistent).

The sensitivity and specificity of using the viral load assays to accurately diagnose HIV infection were calculated using the Roche COBAS AmpliPrep/COBAS TaqMan HIV-1 Qualitative Test, v2.0 assay as this assay is currently the standard test used for clinical management in Uganda. The score-based Wilson method [21] was used to construct confidence intervals for sensitivity and specificity. Confidence intervals for Cohen's Kappa were estimated [22]. McNemar's chi-squared test for symmetry of rows and columns in a two-dimensional contingency table was estimated [23]. Further, a sub-analysis was conducted comparing the performance of the quantitative assay with the qualitative assay in infants exposed to antiretroviral drugs–either through infant prophylaxis or maternal treatment. All statistical analyses were performed in the R statistical computing environment.

This study was approved by the Uganda National Council for Science and Technology; the Higher Degrees, Research and Ethics Committee from Makerere University, Uganda; Chesapeake International Review Board in the United States; and the Ethics Review Committee from the World Health Organization, Geneva, Switzerland. Informed consent was waived by each ethical review committee because of the use of routine, leftover clinical samples. The data were fully anonymized prior to access and analysis. Viral load test results were not provided to patients. The routine clinical qualitative infant diagnosis test results were returned to the health care facility and caregiver per national guidelines.

## Results

A total of 858 infant samples were included in the study, of which half were female (50.6%). The median age of infants tested was seven months, with 33.0% less than three months of age and 35.2% older than nine months of age. Seventy-four percent of all mothers were taking antiretroviral therapy (<10% unknown), including 62.4% of mothers with HIV-infected infants (Table 1). Sixty-eight percent of infants received some form of prophylaxis (12.7% unknown), including 57.9% of HIV-infected infants. A significant proportion (76.8%) of infants were exposed to either maternal treatment or infant prophylaxis, including 66.9% of the HIV-infected infants.

There were 263 HIV-infected and 260 HIV-uninfected infants included in the Roche group and 257 HIV-infected and HIV-uninfected infants in the Abbott group (Table 2). No infant samples were excluded; however, there were 18, 3, and 36 invalid tests or depleted samples using the Roche FVE, Roche SPEX, and Abbott assays, respectively. Because they were unable to provide a valid test result, they were not included in the primary analyses. All patients were tested using Roche COBAS AmpliPrep/COBAS TaqMan HIV-1 Qualitative Test, v2.0 for clinical diagnosis, with the result serving as the reference. The median qualitative cycle threshold value for HIV-infected infants included in the Roche analyses was 24.0 (IQR: 22.3–27.3). Over 20% of HIV-infected infants (56 of 263) had a test result with a qualitative cycle threshold value of 28 or higher and 10% of HIV-infected infants (26 of 263) had a test result with a qualitative cycle threshold value of 30 or higher. The median qualitative cycle threshold value for infants included in the Abbott analysis was 24.1 (IQR: 22.4–26.7). Over 20% of HIV-infected infants (53 of 257) had a test result with a qualitative cycle threshold value of 28 or higher and

**Table 1. Demographic characteristics of study participants.**

| | | Total, N = 858 | Total, N = 338 | Total, N = 520 |
|---|---|---|---|---|
| | | All infants, n (%) | All HIV-uninfected infants, n (%) | All HIV-infected infants, n (%) |
| Gender | | | | |
| | Female | 434 (50.6) | 175 (51.8) | 259 (49.8) |
| | Male | 420 (49.0) | 162 (47.9) | 258 (49.6) |
| Age group | | | | |
| | 0–3 mo | 283 (33.0) | 143 (42.3) | 140 (26.9) |
| | 3–6 mo | 123 (14.3) | 34 (10.1) | 89 (17.1) |
| | 6–9 mo | 150 (17.5) | 51 (15.1) | 99 (19.0) |
| | 9–18 mo | 302 (35.2) | 110 (32.5) | 192 (36.9) |
| Maternal ART | | | | |
| | ART (Option B+) | 310 (36.1) | 147 (43.5) | 163 (31.3) |
| | Option B | 125 (14.6) | 53 (15.7) | 72 (13.8) |
| | Option A | 200 (23.3) | 110 (32.5) | 90 (17.3) |
| | None | 143 (16.7) | 4 (1.2) | 139 (26.7) |
| | Unknown | 80 (9.3) | 24 (7.1) | 56 (10.8) |
| Infant prophylaxis | | | | |
| | Daily NVP through BF | 10 (1.2) | 3 (0.9) | 7 (1.3) |
| | Daily NVP to 6 wks | 505 (58.9) | 274 (81.1) | 231 (44.4) |
| | sdNVP + AZT for 7 days | 20 (2.3) | 2 (0.6) | 18 (3.5) |
| | sdNVP only | 50 (5.8) | 5 (1.5) | 45 (8.7) |
| | None | 164 (19.1) | 15 (4.4) | 149 (28.7) |
| | Unknown | 109 (12.7) | 39 (11.5) | 70 (13.5) |
| Exposed to either maternal ART or infant prophylaxis | | 659 (76.8) | 311 (92.0) | 348 (66.9) |

BF: breastfeeding.

sdNVP: single dose NVP.

11% of HIV-infected infants (29 of 257) had a test results with a qualitative cycle threshold value of 30 or higher.

## Viral load as an infant diagnostic using Roche and FVE protocol for dried blood spot preparation

The sensitivity and specificity of using the Roche viral load assay as a diagnostic with the FVE dried blood spot preparation protocol were 84.7% (95% CI: 79.7–88.6%) and 99.6% (95% CI: 97.8–100%), respectively (Table 3). The kappa coefficient was 0.845 (95% CI: 0.799–0.891). The median quantitative cycle threshold value was 28.6 (IQR: 27.0–30.8), while the median viral load was 18,624 copies/ml (IQR: 5,277–49,935 copies/ml). Using the FVE protocol, there were 37 false negatives, which had a median qualitative cycle threshold value of 30.8 (IQR: 29.3–32.2). One false positive had a quantitative cycle threshold value of 38.2 and a viral load of < 400 copies/ml.

## Viral load as an infant diagnostic using Roche and SPEX buffer for dried blood spot preparation

The sensitivity and specificity of using the Roche viral load assay as a diagnostic with the SPEX dried blood spot preparation protocol were 98.9% (95% CI: 96.7–99.6%) and 98.8% (95% CI:

**Table 2. Demographic characteristics of study participants by technology.**

| | | Roche TaqMan v2 | | | Abbott m2000 | | |
|---|---|---|---|---|---|---|---|
| | | Total, N = 523 | Total, N = 260 | Total, N = 263 | Total, N = 514 | Total, N = 257 | Total, N = 257 |
| | | All infants, n (%) | All HIV-uninfected infants, n (%) | All HIV-infected infants, n (%) | All infants, n (%) | All HIV-uninfected infants, n (%) | All HIV-infected infants, n (%) |
| Gender | | | | | | | |
| | Female | 264 (50.5) | 138 (53.1) | 126 (47.9) | 267 (51.9) | 134 (52.1) | 133 (51.8) |
| | Male | 259 (49.5) | 122 (46.9) | 137 (52.1) | 243 (47.3) | 122 (47.5) | 121 (47.1) |
| Age group | | | | | | | |
| | 0–3 mo | 175 (33.5) | 107 (41.2) | 68 (25.9) | 171 (33.2) | 99 (38.5) | 72 (28.0) |
| | 3–6 mo | 74 (14.2) | 30 (11.5) | 44 (16.7) | 59 (11.5) | 14 (5.4) | 45 (17.5) |
| | 6–9 mo | 97 (18.5) | 35 (13.5) | 62 (23.6) | 81 (15.8) | 44 (17.1) | 37 (14.4) |
| | 9–18 mo | 177 (33.8) | 88 (33.8) | 89 (33.8) | 203 (39.5) | 100 (38.9) | 103 (40.1) |
| Maternal ART | | | | | | | |
| | ART (Option B+) | 207 (39.6) | 126 (48.5) | 81 (30.8) | 187 (36.4) | 105 (40.9) | 82 (31.9) |
| | Option B | 79 (15.1) | 40 (15.4) | 39 (14.8) | 76 (14.8) | 43 (16.7) | 33 (12.8) |
| | Option A | 120 (22.9) | 77 (29.6) | 43 (16.3) | 136 (26.5) | 89 (34.6) | 47 (18.3) |
| | None | 78 (14.9) | 3 (1.2) | 75 (28.5) | 67 (13.0) | 3 (1.2) | 64 (24.9) |
| | Unknown | 39 (7.5) | 14 (5.4) | 25 (9.5) | 48 (9.3) | 17 (6.6) | 31 (12.1) |
| Infant prophylaxis | | | | | | | |
| | Daily NVP through BF | 6 (1.1) | 3 (1.2) | 3 (1.1) | 6 (1.2) | 2 (0.8) | 4 (1.6) |
| | Daily NVP to 6 wks | 335 (64.1) | 215 (82.7) | 120 (45.6) | 320 (62.3) | 209 (81.3) | 111 (43.2) |
| | sdNVP + AZT for 7 days | 13 (2.5) | 2 (0.8) | 11 (4.2) | 9 (1.8) | 2 (0.8) | 7 (2.7) |
| | sdNVP only | 21 (4) | 4 (1.5) | 17 (6.5) | 32 (6.2) | 4 (1.6) | 28 (10.9) |
| | None | 84 (16.1) | 8 (3.1) | 76 (28.9) | 87 (16.9) | 14 (5.4) | 73 (28.4) |
| | Unknown | 64 (12.2) | 28 (10.8) | 36 (13.7) | 60 (11.7) | 26 (10.1) | 34 (13.2) |
| Exposed to either maternal ART or infant prophylaxis | | 421 (80.5) | 244 (93.8) | 177 (67.3) | 408 (79.4) | 237 (92.2) | 171 (66.5) |

BF: breastfeeding.

sdNVP: single dose NVP.

**Table 3. (a) Performance of the Roche viral load test using the FVE protocol compared to the Roche qualitative test.** (b) Performance of the Roche viral load test using SPEX buffer compared to the Roche qualitative test.

| (a) | | Qualitative | | | | |
|---|---|---|---|---|---|---|
| | | Positive | Negative | Sensitivity (95% CI) | Specificity (95% CI) | Cohen Kappa (95% CI) |
| Quantitative | Positive | 211 | 1 | 84.7% (79.7–88.6) | 99.6% (97.8–100) | 0.845 (0.799–0.891) |
| | Negative | 37 | 256 | | | |
| (b) | | Qualitative | | | | |
| | | Positive | Negative | Sensitivity (95% CI) | Specificity (95% CI) | Cohen Kappa (95% CI) |
| Quantitative | Positive | 259 | 3 | 98.9% (96.7–99.6) | 98.8% (96.6–99.6) | 0.977 (0.959–0.995) |
| | Negative | 3 | 255 | | | |

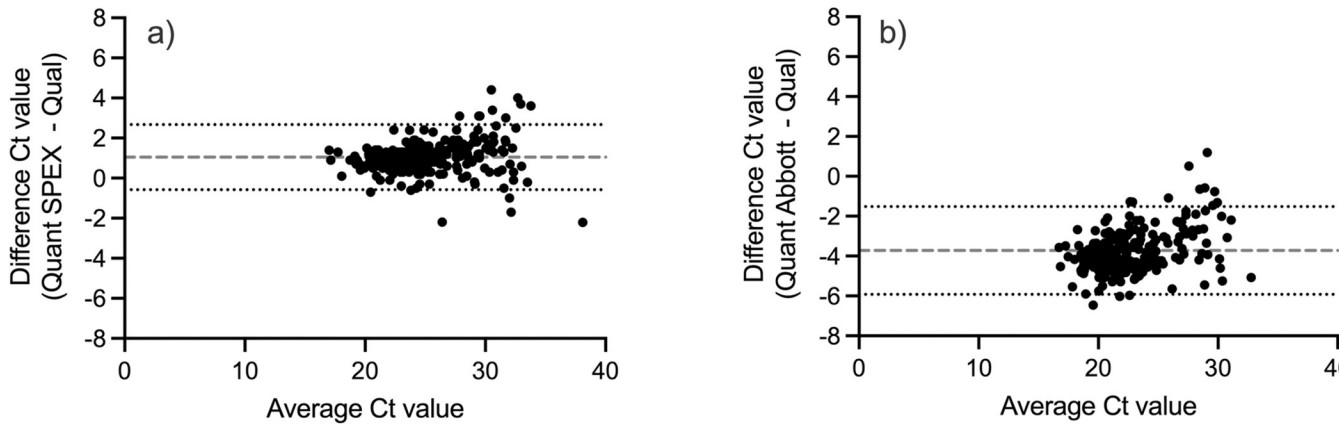

**Fig 1.** Bland-Altman analysis of quantitative Roche SPEX (a) and quantitative Abbott (b) compared to the qualitative assay.

96.6–99.6%), respectively (Table 3) (Fig 1A). The kappa coefficient was 0.977 (95% CI: 0.959–0.995) and McNemar was 1.000. The median quantitative cycle threshold value was 25.0 (IQR: 23.1–28.5), while the median viral load was 224,222 copies/ml (IQR: 26,525–613,020 copies/ml). There were three false negatives with qualitative cycle threshold values of 32.7, 33.1, and 34.0. There were three false positives with quantitative cycle threshold values of 40.2, 33.5, and 41.2 and viral load values all < 400 copies/ml.

As expected, the viral load and cycle threshold quantities of the positive samples were inversely correlated: as the viral load values increased, the cycle threshold values decreased (Fig 2A). Interestingly, the viral load values of the positive samples were consistently lower when using the Roche FVE protocol compared with the Roche SPEX protocol. This was expected given the lower performance of the Roche FVE protocol to accurately detect HIV.

### Viral load as an infant diagnostic using Abbott

The sensitivity and specificity of the Abbott Real*Time* HIV-1 Viral Load assay were 95.2% (95% CI: 91.6–97.3%) and 99.2% (95% CI: 97.1–99.8%), respectively (Table 4) (Fig 1B). The kappa coefficient was 0.946 (95% CI: 0.916–0.975) and McNemar was 0.027. The median quantitative cycle threshold value was 20.1 (IQR: 18.4–22.5), while the median viral load was 464,132 copies/ml (IQR: 894–9,351,630 copies/ml). There were 11 false negatives and two false positives. The 11 false negatives had a median qualitative cycle threshold value of 31.9 (30.4–33.0). The two false positives had quantitative cycle threshold values of 29.2 (<839 copies/ml) and 29.1 (<839 copies/ml). Though there were 11 false negatives, the viral load values inversely correlated with the cycle thresholds and more closely mirrored those with the Roche SPEX protocol (Fig 2B). Additional testing with the Abbott Real*Time* assay using the alternative protocol, in which the samples were processed using the mDBS buffer, produced similar results with no significant improvement in sensitivity (sensitivity: 96.2% (95% CI: 92.8–98.1%) and specificity: 98.6% (95% CI: 95.9–99.5%)).

### Sub-analysis of using viral load as a diagnostic for infants exposed to antiretroviral drugs

A separate analysis reviewed the performance of using viral load as an infant diagnosis assay for infants exposed to antiretroviral drugs–either through infant prophylaxis or maternal treatment. The sensitivity and specificity of using the Roche viral load assay as a diagnostic with the

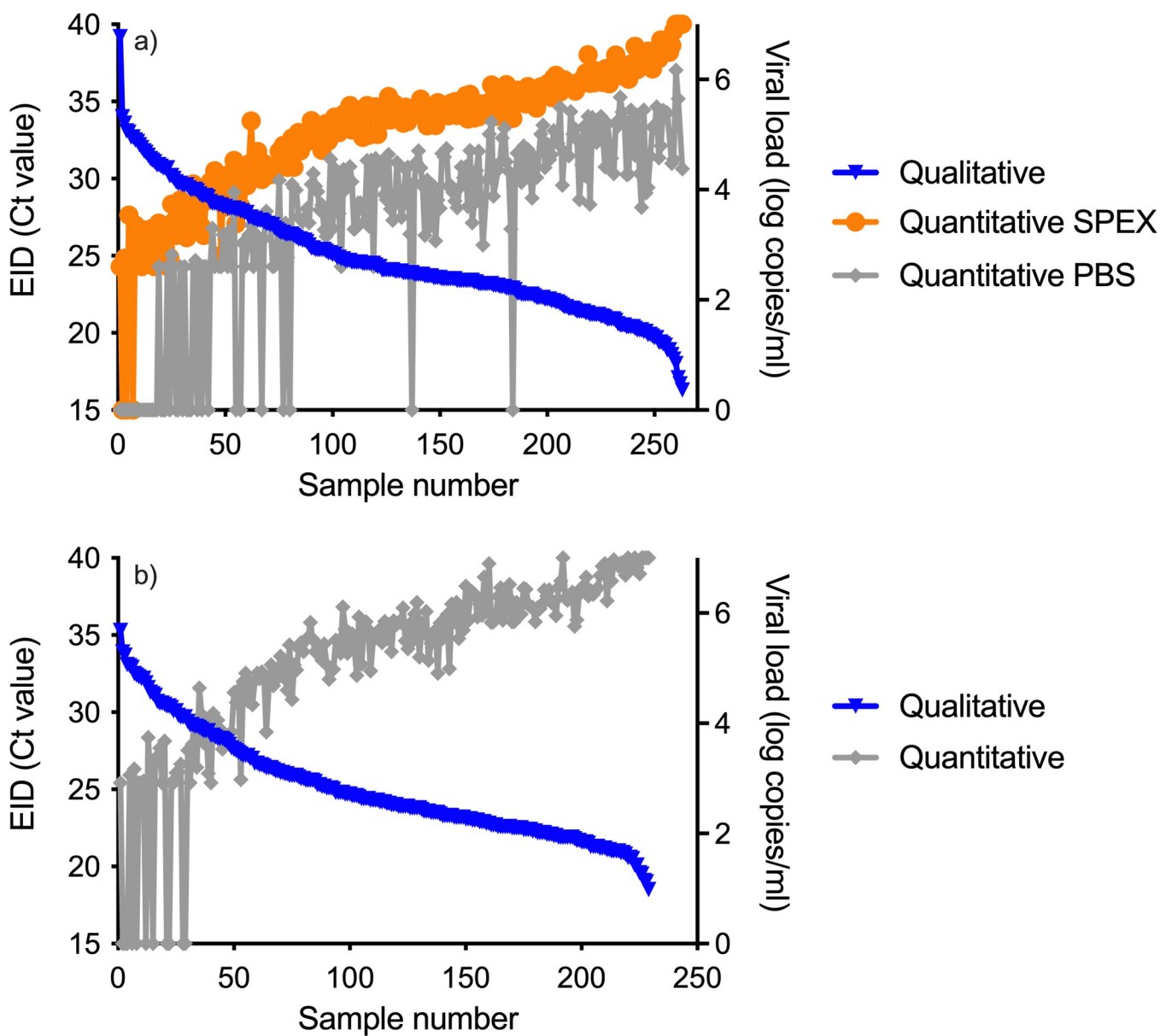

**Fig 2.** Qualitative cycle threshold values of qualitative and quantitative viral loads for both Roche protocols (a) and Abbott (b).

SPEX dried blood spot preparation protocol for only antiretroviral-exposed children were 98.9% (95% CI: 96.0–99.7) and 98.8% (95% CI: 96.4–99.6), respectively. The kappa coefficient was 0.975 (95% CI: 0.954–0.997) and McNemar was 1.000. The sensitivity and specificity of

**Table 4. Performance of the Abbott viral load test compared to the Roche qualitative test.**

|  |  | Qualitative | | Sensitivity | Specificity | Cohen Kappa |
|---|---|---|---|---|---|---|
|  |  | Positive | Negative | (95% CI) | (95% CI) | (95% CI) |
| Quantitative | Positive | 225 | 2 | 95.2% | 99.2% | 0.946 |
|  | Negative | 11 | 249 | (91.6–97.3) | (97.1–99.8) | (0.916–0.975) |

the Abbott Real*Time* HIV-1 Viral Load assay were 94.7% (95% CI: 89.9–97.3%) and 99.6% (95% CI: 97.6–100%), respectively. The kappa coefficient was 0.950 (95% CI: 0.918–0.982) and McNemar was 0.046.

## Discussion

Though previous data were primarily published before 2010 and prior to widespread maternal treatment and infant prophylaxis, these results confirmed that HIV viral load assays can be used to diagnose HIV infection in infants. The performance of both the Roche viral load using SPEX and Abbott viral load tests were above the WHO recommended 95% sensitivity and 98% specificity [15, 16]. Interestingly, the false positive and false negative results had either low viral load levels on the quantitative assay or high cycle threshold levels on the qualitative assay, respectively, using either the Roche SPEX and Abbott assays, warranting further investigation. Had recent WHO recommendations of an indeterminate range been implemented routinely to interpret clinical test results, the false negative results would have been classified as indeterminate by the qualitative assay rather than positive, while all false positive results were detectable yet below the limit of quantification, again likely to be within the indeterminate range [17, 24, 25].

While it is concerning for infants to receive potentially false negative results and be missed, recent studies have shown that 75% of indeterminate test results are negative, and the infant likely negative, upon repeat testing [24]. Furthermore, infants classified as indeterminate should not immediately start treatment, but have the sample repeated to determine whether they are truly infected [17, 24]. These potential false negative results further emphasize the importance of confirmatory testing of all detectable samples as well as the need to strengthen retention and end of exposure testing of all infants with initial negative results.

There was a significant difference observed in the performance of the Roche viral load assay depending on how the dried blood spot was processed. PBS is a water-based salt solution that is minimally invasive to cells as it is isotonic, non-toxic, and non-damaging. Due to its nature, PBS buffer does not lyse the cell membrane and typically only allows for extraction and elution of extracellular RNA. SPEX buffer has been the standard buffer used in the Roche qualitative assay. It is a chaotropic guanidinium-based sample pre-extraction buffer that lyses the cell membrane and provides accessibility to intracellular nucleic acids [19, 26]. Therefore, when processing dried blood spots with SPEX, intracellular RNA and proviral DNA are extracted and eluted for subsequent amplification in addition to circulating extracellular RNA [26, 27]. Understandably, therefore, using PBS buffer in the FVE protocol resulted in significantly lower viral loads and higher cycle thresholds, indicating that less nucleic acids were amplified when compared to both the qualitative and quantitative results using the standard SPEX buffer. Unfortunately, implementing this intervention using the more effective SPEX buffer would result in different dried blood spot sample processing methods for viral load and infant diagnosis samples. Using SPEX buffer to process dried blood spot samples for routine viral load testing results in significant over-quantification compared to plasma and is generally not suggested [16, 28]. However, there are several different sample processing modalities (ie. plasma, plasma separation card, whole blood, dried blood spot) that may need to be considered and analyzed for both test types and across settings, while testing and viral load reagents can remain the same.

Countries are currently using one or multiple sample types for viral load testing of people living with HIV. In several countries, both plasma and dried blood spots are used. In health care facilities or countries where plasma is the primary sample type for viral load testing, this intervention of using viral load as a diagnostic for infants could still be considered. At the

health care level, dried blood spots could still be provided and utilized for the infant population; however, once the sample is delivered to and processed by the laboratory, follow-up activities would follow the viral load testing process, utilizing viral load reagents in particular. Similar slight adaptations may be the case in settings where other sample collection kits are used, such as dried plasma spots, plasma separation cards, or dried blood spots processed using FVE. Otherwise, in settings where dried blood spot samples are used for viral load testing of people living with HIV, commodities and processes used throughout the sample collection and testing process could be streamlined for infant testing.

Though often referred to as DNA PCR (polymerase chain reaction), it is important to note that all infant diagnosis assays currently on the market in low- and middle-income countries are RNA-specific or detect total nucleic acids (TNA) [27, 29]. For example, despite being RNA-specific the Abbott m-PIMA Detect infant diagnosis assay has high sensitivity and specificity compared to laboratory-based technologies [28, 30]. The requirement or suggestion for DNA-specific infant diagnosis assays should, therefore, be reconsidered and the terminology for infant diagnosis technologies simply noted as PCR (polymerase chain reaction) or NAT (nucleic acid technology) rather than the incorrect 'DNA PCR' nomenclature. As well, molecular tests used with whole blood (including dried blood spots) often extract, detect, and amplify greater and significant quantities of RNA, both intracellular and extracellular, compared to DNA [27]. Nucleic acid extraction methods are not always nucleic acid discriminatory; therefore, using whole blood samples will often allow for detection of intracellular DNA and RNA in addition to extracellular RNA [27].

Several challenges remain in Uganda and globally to ensure and increase timely access to infant diagnosis. Variable pricing between infant diagnosis and viral load test kits, reagent stock outs, duplicative workflows, and sample batching are examples of issues that plague national infant diagnosis programs. Creating a more efficient laboratory system through consideration of viral load assays as a diagnostic may support lower reagent prices, reduced stock outs since viral load reagents are generally more available due to high volumes and consistent utilization, and streamlined workflows due to the ability to integrate testing. Implementation studies and/or cost-benefit analyses may further support this novel intervention. Though not currently recommended, using viral load as a diagnostic would also provide the laboratory and clinician with a clinical viral load test result. Finally, within current and/or pipeline viral load assays, manufacturers would ideally adjust or develop appropriately considered intended use claims (often termed a 'dual claim') and seek regulatory approvals–these would include defining the limits of detection and appropriate indeterminate ranges. Doing so would similarly create efficiencies for manufacturers in having fewer products to manage and produce on similar manufacturing lines.

Interestingly, over 60% of infants with a positive infant diagnosis were exposed to maternal antiretrovirals and nearly 60% of infants with a positive infant diagnosis received some infant prophylaxis. In total, 67% of infants with a positive infant diagnosis were exposed to either maternal or infant antiretrovirals, while 92% of infants with a negative infant diagnosis were exposed to either maternal or infant antiretrovirals. These results would suggest challenges in the PMTCT cascade, primarily that PMTCT access should continue to be expanded and that some mothers enrolled in PMTCT may potentially be intermittently taking treatment during pregnancy and/or breastfeeding, accessing antiretroviral therapy late in pregnancy or breastfeeding, receiving sub-optimal treatment regimens or transmitting drug resistant virus [31–35].

Results from this study are relevant and generalizable to other LMIC settings. The majority of mothers were on antiretroviral therapy (>75%), while most infants were receiving prophylaxis. However, these rates are suboptimal and continued programmatic scale-up is essential.

Furthermore, some infants had low levels of viremia or high cycle threshold counts, as expected in improved PMTCT programs [36]. Furthermore, HIV-infected infants were significantly older than those who were HIV-uninfected (7.5 months versus 6.0 months, p<0.001). This has been observed elsewhere [7, 8] and would highlight the need to strengthen case-finding strategies to test and identify HIV-infected infants earlier to prevent early morbidity and mortality [37, 38]. Even with these relatively high rates of treatment exposure and low levels of viremia, the viral load quantitative assays were able to successfully detect positive samples. However, the importance of confirmatory testing remains critical as the prevalence of mother to child transmission and levels of viremia reduce [17, 24–25, 36, 39]. Finally, testing through the entire infant diagnosis cascade until the end of the exposure period is critical to identify all HIV-infected infants, particularly as the improvement of PMTCT programs has resulted in infants now being more likely to become infected during the breastfeeding period than *in utero* [3, 40].

There were several limitations in this study. Sample sizes within the highest cycle threshold or lower viral load values were limited. However, the overall sample size was large and confidence intervals were well within +/-5%. Furthermore, the sampling technique ensured that the population included in this study was highly generalizable in similar LMIC settings. Most mothers and infants were receiving antiretroviral treatment or prophylaxis, respectively, and in similar proportions compared to the PMTCT coverage rate estimated in Uganda and the region [2, 3]. There are concerns that exposure to antiretroviral drugs could reduce the performance of quantitative assays when used as a diagnostic due to lower viral load levels; however, sub-analyses in that population indicated comparable performance. Though the majority of specimens were obtained from infants and young children who were exposed to ARVs, the median age at testing was seven months and the majority of infants were tested after three months of age–well after the standard six-week test time, the age up until HIV-exposed infants are provided daily nevirapine. Ideally additional technologies, such as the Abbott mPIMA, Cepheid GeneXpert and Hologic Aptima would also be studied; however, the present study provides a clear proof of principle and confirmation that viral load tests can be used as a diagnostic within current programmatic settings in high HIV burden countries. Policy adoption and early implementation should be considered across settings to maximize resources, streamline laboratory systems, and provide greater access to testing.

## Conclusions

In order to achieve ambitious global targets, particularly for improving access to infant HIV testing and treatment, which has stagnated in recent years, creative, new innovations are critical. This study demonstrates the potential for HIV viral load quantitative assays to be used to diagnose HIV infection in infants under 18 months of age. Considering this integrated approach may lead to more efficient and streamlined systems, both for national programs, within the laboratory, and through procurement, as well as for manufacturers.

## Acknowledgments

We would like to gratefully acknowledge the laboratory staff at the Central Public Health Laboratory.

## Author Contributions

**Conceptualization:** Charles Kiyaga, Christopher Okiira, Trevor Peter, Meg Doherty, Jilian A. Sacks, Lara Vojnov.

**Data curation:** Youyi Fong, Grace Esther Kushemererwa, Isaac Ssewanyana, Lara Vojnov.

**Formal analysis:** Youyi Fong, Lara Vojnov.

**Investigation:** Youyi Fong, Christopher Okiira, Ismail Kayongo, Iga Tadeo, Christine Namulindwa, Victor Bigira, Lara Vojnov.

**Methodology:** Youyi Fong, Christopher Okiira, Grace Esther Kushemererwa, Christine Namulindwa, Victor Bigira, Isaac Ssewanyana, Trevor Peter, Jilian A. Sacks, Lara Vojnov.

**Project administration:** Charles Kiyaga.

**Supervision:** Charles Kiyaga, Christopher Okiira, Grace Esther Kushemererwa, Ismail Kayongo, Victor Bigira, Isaac Ssewanyana, Meg Doherty, Lara Vojnov.

**Validation:** Grace Esther Kushemererwa, Ismail Kayongo, Iga Tadeo, Christine Namulindwa, Victor Bigira.

**Writing – original draft:** Lara Vojnov.

**Writing – review & editing:** Charles Kiyaga, Youyi Fong, Christopher Okiira, Grace Esther Kushemererwa, Ismail Kayongo, Iga Tadeo, Christine Namulindwa, Victor Bigira, Isaac Ssewanyana, Trevor Peter, Meg Doherty, Jilian A. Sacks.

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
