## [Decision Letter · Decision Letter 0]

23 Jul 2021

PONE-D-21-12685

Viral load assays when used with whole blood perform well as a diagnostic for infants

PLOS ONE

Dear Dr. Vojnov,

Thank you for submitting your manuscript to PLOS ONE. After careful consideration, we feel that it has merit but does not fully meet PLOS ONE’s publication criteria as it currently stands. Therefore, we invite you to submit a revised version of the manuscript that addresses the points raised during the review process.

Most of the issues are minor and we are committed to a rapid turnaround of the revision once submitted. Please pay particular attention to describe the source population and age of infant tested and consider the issues raised by the reviewer. 

We look forward to receiving your revised manuscript.

Kind regards,

Nei-yuan Hsiao

Academic Editor

PLOS ONE

Additional Editor Comments (if provided):

I would like to see a separate analysis in result and/or discussion around the sen/spec of VL assays in the sub-population of ARV exposed children. This will be particular relevant as more countries increase the maternal ART coverage and infant prophylaxis.

Reviewers' comments:

Reviewer's Responses to Questions

**Comments to the Author**

1. Is the manuscript technically sound, and do the data support the conclusions?

Reviewer #1: Yes

Reviewer #2: Yes

2. Has the statistical analysis been performed appropriately and rigorously? 

Reviewer #1: Yes

Reviewer #2: Yes

3. Have the authors made all data underlying the findings in their manuscript fully available?

Reviewer #1: No

Reviewer #2: Yes

4. Is the manuscript presented in an intelligible fashion and written in standard English?

Reviewer #1: Yes

Reviewer #2: Yes

5. Review Comments to the Author

Reviewer #1: PONE-D-21-12685 Review

Line 2 & 22 – specify in the title that it is HIV viral load. This should also be corrected in the discussion, for completeness.

Why does the title speak of whole blood when the research was done on dried blood spots?

My suggestion for the title of the paper:

A laboratory validation of commercial viral load assays for use in early infant diagnosis of HIV

Line 32 & 47 – the word “diagnostic” is confusing to the reader here, as it is an adjective being used as a noun. Do you mean that you are using a quantitative assay, with a qualitative interpretation of the result? Perhaps you should replace the term with the more widely-used term “Early Infant Diagnosis (EID) of HIV” which pertains to the qualitative diagnosis of HIV in children versus quantitative monitoring of HIV infection – see line 226 below.

Have these commercial methods not already been validated? What is different about your study, or the added benefit? Is t a field evaluation? Or a validation of a deviation from the commercial method, e.g. change in specimen type? This is unclear to the reader in the last line of the objective.

Line 73 – Reference 11 reported that the qualitative NASBA was superior to the quantitative for infant diagnosis

Line 78 – Incorrect terminology - “detection efficiency” should be “ accuracy”

Line 84 – Suggestion: “to prioritize qualitative HIV DNA assays for EID; however…”

Line 90 – Suggestion: “to diagnose HIV infection in infants by applying a qualitative interpretation of results”

Line 106 – Suggestion: “70 additional consecutive negative samples were tested using only the Abbott viral load assay, as the original samples were insufficient for testing with both assays.”

Line 226 Suggestion: “…these results confirmed that viral load assays can be used for Early Infant Diagnosis (EID) of HIV.”

Line 353 - the link doesn’t work

It would be good to have some clinical information on those patients in whom the results were categorized as false negative or false positive, in order to get some idea of where or why the method fails.

It would be good to plot Ct values of both methods against each other in a Bland Altman plot, in order to ascertain subtle bias etc.

Reviewer #2: Thank you for the opportunity to review this article. The article is well written and answers an important question with regards to HIV diagnostic modalities for infants and young children - namely, the performance of quantitative virological assays for qualitative diagnosis using dried blood spots specimens. I having the following comments/suggestions:

Abstract

- Include time period of sample collection and age of participants in the abstract (otherwise it's very abstract)

Methods

- Suggest including the specific type of nucleic acid detected by each assay evaluated (i.e. total nucleic acid by CAP/CTM; RNA only by Abbott RealTime)

Results

- Include the number and proportion of invalid results using each method

Discussion

- Line 233-234, 'while all false positive results were detectable yet below the limit of detection' - I think you mean below the lower limit of quantification?

- The limitations section should be expanded to include mention of the age range of study participants. Although the majority of specimens were obtained from infants and young children who were exposed to ARVs (including maternal ART and infant prophylaxis), they would not have been exposed to infant prophylactic regimens at time of specimen collection (as the majority were exposed to daily NVP for 6 weeks but the majority of samples were taken >3mo of age). Hence, the performance of the assays were not specifically evaluated at a time when the routine early infant diagnosis is performed (i.e. around 6 weeks of age). This is a major limitation and needs to be explicitly acknowledged, especially considering viraemia is found to be lower among younger infected infants. This could impact on lower diagnostic sensitivity of quantitative virological assays.

6. PLOS authors have the option to publish the peer review history of their article (what does this mean?). If published, this will include your full peer review and any attached files.

Reviewer #1: No

Reviewer #2: **Yes: **Ahmad Haeri Mazanderani

---

## [Author Response · Author response to Decision Letter 0]

14 Feb 2022

Response to Reviewers

Editor’s comments

1. Please provide additional details regarding participant consent. In the ethics statement in the Methods and online submission information, please ensure that you have specified (1) whether consent was informed and (2) what type you obtained (for instance, written or verbal, and if verbal, how it was documented and witnessed). If your study included minors, state whether you obtained consent from parents or guardians. If the need for consent was waived by the ethics committee, please include this information.

Response: This has been updated in the Methods (lines 153-155) as well is in the online submission.

2. For your uploaded Response to Reviewers which include a point by point response to each of the points made by the Editor and / or Reviewers, please change the file type as a 'Response to Reviewers'.

Please follow this link for more information: http://blogs.PLOS.org/everyone/2011/05/10/how-to-submit-your-revised-manuscript/

Response: this has been done accordingly.

3. It also is important that you include a cover letter with your manuscript. Please ensure that this letter is addressed specifically to PLoS ONE. Please also include:

* why this manuscript is suitable for publication in PLoS ONE.

* how does your paper provide a worthwhile addition to the scientific literature?

* how does your paper relate to previously published work? 

* which types of scientists do you believe will be most interested in your study?

Response: We have revised our cover letter accordingly.

3. We note your Data Availability statement as follows: "Most of the data is contained within the manuscript. National policies and ethics in Uganda do not allow for the full data set to be provided online or as Supporting Information."

Please also provide non-author contact information* for a data access committee, ethics committee, or other institutional body to which data requests may be sent.

Response: Certainly, please see here. Mr Ronald Jjagwe at Uganda National Council for Science and Technology, info@uncst.go.ug.

I would like to see a separate analysis in result and/or discussion around the sen/spec of VL assays in the sub-population of ARV exposed children. This will be particular relevant as more countries increase the maternal ART coverage and infant prophylaxis.

Response: we have updated the work to do such a sub-analysis and included it within the Methods (lines 146-148), Results (230-239), and Discussion (348-354).

Reviewer #1: PONE-D-21-12685 Review

Line 2 & 22 – specify in the title that it is HIV viral load. This should also be corrected in the discussion, for completeness.

Why does the title speak of whole blood when the research was done on dried blood spots?

My suggestion for the title of the paper:

A laboratory validation of commercial viral load assays for use in early infant diagnosis of HIV

Response: We have adjusted the title to include ‘HIV viral load’ and throughout the document for clarity. The title and body of text speaks to whole blood interchangeably with dried blood spots because the latter are a whole blood specimen. These aren’t really different specimen types. Also, this wasn’t a ‘validation’ as defined. Further, global language is moving away from using the terminology ‘early infant diagnosis (or EID)’ as much of the testing being conducted is beyond the early time point (within the first two months of life) and to ensure/encourage testing continue to happen throughout and after the period of exposure – a current significant gap. Several recent WHO documents have been published with this new language.

Line 32 & 47 – the word “diagnostic” is confusing to the reader here, as it is an adjective being used as a noun. Do you mean that you are using a quantitative assay, with a qualitative interpretation of the result? Perhaps you should replace the term with the more widely-used term “Early Infant Diagnosis (EID) of HIV” which pertains to the qualitative diagnosis of HIV in children versus quantitative monitoring of HIV infection – see line 226 below.

Response: We have attempted to revise this within the title but also throughout the manuscript – as a ‘diagnostic assay’. As for ‘EID’, please see the response above.

Have these commercial methods not already been validated? What is different about your study, or the added benefit? Is t a field evaluation? Or a validation of a deviation from the commercial method, e.g. change in specimen type? This is unclear to the reader in the last line of the objective.

Response: Correct, these methods have not been validated with most suppliers so a deviation from the commercial method in an effort a) to encourage suppliers to update their claims; and b) provide evidence for guidance and policy changes. The paragraph starting on line 303 attempts to highlight and explain this.

Line 73 – Reference 11 reported that the qualitative NASBA was superior to the quantitative for infant diagnosis

Response: Upon additional review of this manuscript, the authors note that qualitative NASBA was superior to (qualitative) DNA PCR, but comparable to quantitative PCR.

Line 78 – Incorrect terminology - “detection efficiency” should be “ accuracy”

Response: This has been revised accordingly (line 80).

Line 84 – Suggestion: “to prioritize qualitative HIV DNA assays for EID; however…”

Response: This has been revised accordingly (line 85).

Line 90 – Suggestion: “to diagnose HIV infection in infants by applying a qualitative interpretation of results”

Response: This has been revised accordingly (line 91).

Line 106 – Suggestion: “70 additional consecutive negative samples were tested using only the Abbott viral load assay, as the original samples were insufficient for testing with both assays.”

Response: This has been revised accordingly (line 109).

Line 226 Suggestion: “…these results confirmed that viral load assays can be used for Early Infant Diagnosis (EID) of HIV.”

Response: This has been revised (line 243).

Line 353 - the link doesn’t work

Response: The link has been updated (line 376).

It would be good to have some clinical information on those patients in whom the results were categorized as false negative or false positive, in order to get some idea of where or why the method fails.

Response: Unfortunately beyond the data presented within, additional clinical information is not accessible. Samples were obtained from routine clinical testing with limited clinical information from requisition forms captured and reported (ie. infant ARV prophylaxis).

It would be good to plot Ct values of both methods against each other in a Bland Altman plot, in order to ascertain subtle bias etc.

Response: This has been included as a new Figure 1 (lines 203-204).

Reviewer #2: Thank you for the opportunity to review this article. The article is well written and answers an important question with regards to HIV diagnostic modalities for infants and young children - namely, the performance of quantitative virological assays for qualitative diagnosis using dried blood spots specimens. I having the following comments/suggestions:

Abstract

- Include time period of sample collection and age of participants in the abstract (otherwise it's very abstract)

Response: This has been included as suggested (lines 34-36).

Methods

- Suggest including the specific type of nucleic acid detected by each assay evaluated (i.e. total nucleic acid by CAP/CTM; RNA only by Abbott RealTime)

Response: This has been included accordingly (lines 101-102 and 107).

Results

- Include the number and proportion of invalid results using each method

Response: This has been included already on line 169-170.

Discussion

- Line 233-234, 'while all false positive results were detectable yet below the limit of detection' - I think you mean below the lower limit of quantification?

Response: This has been adjusted accordingly (line 251).

- The limitations section should be expanded to include mention of the age range of study participants. Although the majority of specimens were obtained from infants and young children who were exposed to ARVs (including maternal ART and infant prophylaxis), they would not have been exposed to infant prophylactic regimens at time of specimen collection (as the majority were exposed to daily NVP for 6 weeks but the majority of samples were taken >3mo of age). Hence, the performance of the assays were not specifically evaluated at a time when the routine early infant diagnosis is performed (i.e. around 6 weeks of age). This is a major limitation and needs to be explicitly acknowledged, especially considering viraemia is found to be lower among younger infected infants. This could impact on lower diagnostic sensitivity of quantitative virological assays.

Response: This has been elaborated on and included within the Discussion (lines 348-354).

---

## [Editor Report · Decision Letter 1]

25 Apr 2022

HIV viral load assays when used with whole blood perform well as a diagnostic assay for infants

PONE-D-21-12685R1

Dear Dr. Vojnov,

We’re pleased to inform you that your manuscript has been judged scientifically suitable for publication and will be formally accepted for publication once it meets all outstanding technical requirements.

Kind regards,

Nei-yuan Hsiao

Academic Editor

PLOS ONE
---

## [Editor Report · Acceptance letter]

22 Jun 2022

PONE-D-21-12685R1 

HIV viral load assays when used with whole blood perform well as a diagnostic assay for infants 

Dear Dr. Vojnov:

I'm pleased to inform you that your manuscript has been deemed suitable for publication in PLOS ONE. Congratulations! Your manuscript is now with our production department. 

Kind regards, 

on behalf of

Dr. Nei-yuan Hsiao 

Academic Editor

PLOS ONE